# How the Outliers Influence the Quality of Clustering?

**DOI:** 10.3390/e24070917

**Published:** 2022-06-30

**Authors:** Agnieszka Nowak-Brzezińska, Igor Gaibei

**Affiliations:** Institute of Computer Science, Faculty of Science and Technology, University of Silesia, Bankowa 12, 40-007 Katowice, Poland; igor.vebster@gmail.com

**Keywords:** clustering, outlier detection, clustering quality indexes, AHC, k-Means

## Abstract

In this article, we evaluate the efficiency and performance of two clustering algorithms: AHC (Agglomerative Hierarchical Clustering) and K−Means. We are aware that there are various linkage options and distance measures that influence the clustering results. We assess the quality of clustering using the Davies–Bouldin and Dunn cluster validity indexes. The main contribution of this research is to verify whether the quality of clusters without outliers is higher than those with outliers in the data. To do this, we compare and analyze outlier detection algorithms depending on the applied clustering algorithm. In our research, we use and compare the LOF (Local Outlier Factor) and COF (Connectivity-based Outlier Factor) algorithms for detecting outliers before and after removing 1%, 5%, and 10% of outliers. Next, we analyze how the quality of clustering has improved. In the experiments, three real data sets were used with a different number of instances.

## 1. Introduction

Data clustering is one of the most effective tools for dealing with large amounts of data [1]. When there is a lot of data, we cannot manage it or extract valuable knowledge. By creating clusters of similar data in large data sets, we naturally divide them into homogeneous groups, which allows us to quickly search for groups of objects best suited to what we are currently looking for. When we have an extensive database of fingerprint images and try to classify a currently examined fingerprint, we have to browse through large amounts of data (a complete review of the entire repository) to find the most suitable image. The idea of clustering assumes that we first analyze the similarity of objects and then combine the most similar objects into groups. These groups contain representatives reflecting the group’s content. Then, in the search process, it is enough to browse the representatives of the groups to find a group most similar to the information sought. Furthermore, only a selected group is analyzed. Whenever the real data is analyzed, creating a good quality group is not always possible. Consequently, it can threaten the effectiveness of searching for information in the group structure. Data outliers are factors that hinder the creation of coherent and separable clusters. That is why the problem of outliers in the data is so significant. In our research, we check how the outliers in data affect the difficulties in creating cohesive and well-separated groups. We use the methods known in the literature to assess the quality of clusters. Therefore, we can compare the quality of clusters containing outliers with the quality of clusters after a prior removal or omission outliers. We expect that the measured quality of the cluster should improve after the outliers have been removed. So, after removing or omitting outliers, the cluster should be more consistent internally and well-separated externally. In other words, the similarity of objects within groups should be even more significant (than before removing outliers). In contrast, the similarity of the groups relative to each other should be small. We should remember that when we collect large amounts of data, one of the most valuable techniques is dividing data into consistent groups and analyzing the created groups. We use two popular clustering algorithms: AHC (a hierarchical type) and K−Means (a partitional type). Both of them are easy to understand and implement. A result is a set of data clusters. These clusters are searched to find accurate information, for example, a given data that matches a query in the best way. As we search only within the representatives of these clusters, we may omit the most relevant data even if they exist in this data set. That is why it is essential to verify the quality of the clusters [2,3]. There are many possible quality indexes for measuring the quality of created clusters. We decided to use the two most popular: *Dunn* and *Davies–Bouldin* indexes.

Similar studies are, of course, carried out by scientists around the world. We decided to test two clustering algorithms with different parameters because, as we know, they always have a significant impact on clustering quality. Our goal is to check which parameters impact a better quality of the created clusters. We also want to check which method allows outliers to be detected. We want to know whether a type of input data affects the effectiveness of detecting outliers and improves the quality of clusters after removing outliers. Does the character of the input data influence the effectiveness of clustering or outlier detection processes? Do the clustering parameters impact the quality of the clusters? Finally, we will confirm that the more outliers we detect, the more the quality of the clusters improves.

The structure of the paper is as follows. Section 2 contains an analysis of the existing knowledge related to the problems of clustering algorithms, methods of detecting outliers, and their impact on the quality of the created clusters. This section contains both the references to the research analyzed by the authors devoted to those particular issues by other scientists and the references to the authors’ works on this topic. Section 3 describes the most important aspects (the definition of clustering, distance measures, and the quality indexes) of the analyzed clustering methods. Outlier detection algorithms (LOF (Local Outlier Factor) and COF (Connectivity-based Outlier Factor)) are introduced in Section 4. The essential research value of the paper is presented in Section 5 containing the research methodology, the description of the used data set, and the results of experiments. It also describes the programming environment used to implement the selected algorithms and the planned experiments. The paper winds up with a summary containing the interpretation of the retrieved results.

## 2. State of the Art

In the literature, one can find a lot of papers on either the comparison of the k−Means and the AHC algorithm, the use of different distance measures or methods of combining clusters, methods of detecting outliers, or, finally, methods of analyzing the quality of clustering. In [4], the authors discuss and compare clustering algorithms and methods of cluster quality assessment (F-measure, Entropy) for different values of the number of clusters. However, they do not investigate the influence of outliers on the clustering results. In [5], the authors compared the clustering times for AHC and k−Means. However, their research does not cover the existence of outliers in the data or the study of the quality of clusters. The authors of [6] present the comparison of the k−Means and AHC algorithms in terms of the number of clusters, the number of objects in clusters, the number of iterations, and clustering times for small and large data. However, the impact of outliers on clustering results or cluster quality research is not included there. A very interesting study was carried out in the paper [7], which compared various clustering algorithms with respect to the size of the data, noise resistance in the data, data types, or the number of input parameters. However, the research in the searched range, i.e., the impact of outliers on the quality of clustering, was not included. The paper [8] presents, in turn, a comparison of dozens of different approaches based on clustering and outlier detection but without any research details. Although, it is impossible to find papers that combine these issues into one study. In [9], we compared the clustering algorithms, outlier detection algorithms, and the methods for assessing the quality of created clusters, but wenever before merged all the issues in one study. We wanted to investigate whether the clustering algorithm we chose (AHC or k−Means algorithm) influences the efficiency of data clusters containing outliers. Moreover, we wanted to find out whether the clustering parameters impact the obtained results.

## 3. Clustering Data Containing Outliers

It is known that clustering algorithms are designed to find objects similar to each other and put them into groups [1]. The more similar objects are, the easier it is to create a group from them. However, it is crucial that in the data we cluster, there is a part of the objects very similar to each other but simultaneously not similar to objects from other groups. If such a condition is met, we receive clusters consistent internally and well-separable externally. Such structures have high quality, most commonly assessed by measures of an internal cohesion assessment (the smallest possible distances within clusters) and an external separation (the highest possible distance between clusters). If outliers appear in the data, they significantly deteriorate the quality of clusters. It is worth emphasizing at this point that the outlier can be both a given error or information noise and real outlier data. Of course, we would like to eliminate these possible data errors just at the stage of data preprocessing because they do not contribute any information to the system and even disturb the created consistency of groups. In turn, the rare data, in reality, can bring a piece of significant, new knowledge to the system, and hence they should not be deleted or combined with all data because we may not see them. Such data should be distinguishable and further analyzed. That is why we propose to discover the outliers before the clustering process. For further analysis, such outliers should be introduced to domain experts, and the clustering process should proceed without outliers. Only then the searching within clusters is efficient.

### 3.1. Clustering Definition and Distance Measures

The main idea of the clustering process is to assign the objects to the created clusters considering their distance or similarity. The greater the distance, the less similar to each other the objects are, and thus they should not belong to the same group. Good-quality clustering requires the created groups to be as internally homogenous and externally distinct as possible. Using a proper distance or similarity measure by a given data type (quantitative, qualitative, or binary) is essential. There are many available measures of the distance or the similarity of data. We can distinguish between measures typically dedicated to numerical data (e.g., Euclidean) and typically linked to categorical data (e.g., Simple Matching Coefficient). In this paper, we use, analyze and compare the *Euclidean* and *Chebyshev* distance measures because we analyze numerical data in the experiments. Having two objects *x* and *y* in a *p*-th multidimensional space (i=1,2,…,p), the distance between these objects can be determined as *Euclidean* (Equation (Equation 1)) or *Chebyshev* (Equation (Equation 2)).
(1)$$d\left(x,y\right)=\sqrt{\sum_{i=1}^{p}{\left(x_{i}-y_{i}\right)}^{2}}$$
(2)$$d\left(x,y\right)=\max_{i}\left(\left|\begin{array}{c} x_{i}-y_{i} \end{array}\right|\right)$$

The choice of these measures has an impact on the results obtained. In the conducted experiments, the distance measure we have chosen in regard to clustering or detecting outliers will further influence whether we will achieve a better quality of the groups.

### 3.2. Clustering Algorithms: Hierarchical vs. Partitional

One of the most general classifications of clustering algorithms defines hierarchical and non-hierarchical clustering algorithms. Hierarchical clustering creates a tree of clusters by identifying and merging similar objects. The primary purpose of hierarchical clustering is to cluster such similar objects.

We used the agglomerative hierarchical clustering (AHC) algorithm. Given a *D* data set of *N* instances, this algorithm (AHC), recursively merges two clusters at each step until all instances are merged into one cluster. The conventional procedure of the AHC, also called the stored dissimilarities approach, takes a pairwise dissimilarity *D* matrix of an *N* size as input, initializes a binary tree with *N* leaves (singletons) with null height values, and iteratively adds new nodes (merged clusters) by fusing a pair of clusters $\left(C_{i},C_{j}\right)$ determined as follows:(3)$$\left(C_{i},C_{j}\right)=\underset{x}{\arg\min}D\left(C_{k},C_{l}\right)$$

AHC can be computationally costly. For the usual AHC procedure described above, the time complexity is $O(N^{3})$. In AHC we use the *Lance–Williams* (LW) formula to calculate the dissimilarity between the initial cluster and a cluster formed by joining two other clusters [10].

We compute the distance between pairs of clusters using the following popular methods: Single Linkage (SL), Complete Linkage (CL), and Average Linkage (AL) [11]. In SL, the distance between two clusters is computed as the shortest possible distance between two points in the clusters, in CL as the distance between two data points furthest apart belonging to different clusters, respectively. At the same time, the AL uses an average distance between each point in the first and the second cluster. The AHC algorithm works as follows:In the first step, each object constitutes a separate cluster. So there are k=N clusters, and we must calculate the distance between each pair of points.Find and join the two most similar clusters reducing the number of clusters by one.Repeat the second step until obtaining the declared final number of clusters (*k*) or combining all objects into one big cluster.

The clustering process should be interrupted when the halted an optimal number of clusters is reached so that their quality is highest.

In each iteration of the K−Means algorithm, we try to divide *N* objects into *k* groups so well that each object belongs to the group to which it is most similar. That division, however, means that if an object in the data set is essentially dissimilar to any cluster, the algorithm will try to include it in one and as a result breaking its internal consistency. Each cluster contains a representative. The representative plays a significant role because the object we want to include in the group is compared with it. Let us suppose it turns out that the distance of a given object from this particular representative is the smallest compared to the distances to the representatives of other groups. In that case, we include this object in this particular group. Then, this object also participates in forming a group representative.

The main idea of the algorithm is as follows:Select the number of clusters (*k*) and assign *k* hypothetical centers at random;For each observation of the original set, a nearest cluster center is determined;The centroids are calculated-these are vectors, the components representing the average values of the particular features, calculated overall records of the cluster;The center of the cluster is shifted to its centroid. Then, the centroid becomes the center of the new cluster;Steps 2–4 are repeated iteratively;The algorithm ends when no cluster changes occur at some iteration.

When analyzing both algorithms, we notice that the AHC algorithm seems more resistant to outliers. Unfortunately, this algorithm, in turn, requires more memory occupation. However, everyone can deal with outliers by seeing that they are those objects in the data set that do not match the created clusters, making them challenging to form.

### 3.3. Clustering Quality Indexes

Cluster validity is a way of assessing the quality of clustering results [2]. We obtain a different partition of the data into groups for the two different clustering algorithms or the same clustering algorithm but with different parameters. We use the cluster quality indexes without knowing which partition is the best. In this work, the *Dunn* and *Davies–Bouldin* indexes were used to validate the clustering quality. The *Dunn* index defines compact groups of clusters, the objects of which are well-grouped together, and the clusters themselves are located as far away from each other as possible. Higher values of the *Dunn* index indicate good quality of clustering-the higher the index, the better [3]. The *Dunn* index for *k* clusters is defined as Equation (Equation 4).
(4)$$D\left(u\right)=\min_{1\leq i\leq k}\left\{\min_{1\leq j\leq k,j\neq i}\left\{\frac{(\delta \left(X_{i},X_{j}\right)}{\max_{1\leq c\leq k}\left\{\Delta \left(X_{c}\right)\right\}}\right\}\right\}$$
where $\delta (X_{i},X_{j})$ is an inter-cluster distance between cluster centroids $C_{i}$ and $C_{j}$, and $\Delta (X_{c})$ is an intra-cluster distance of cluster $X_{c}$. The indexes itself is sensitive to noise and outliers in the data. The index’s modifications reduce this error with different methods of measuring inter-cluster distance. The quality of clustering performed using the quantitative and qualitative characteristics of the data set is shown by the *Davies–Bouldin* index. Since clusters must be compact and well-separated, the lowest possible index value indicates high-quality clustering. The *Davies–Bouldin* index for *k* clusters is defined as Equation (Equation 5).
(5)$$DB\left(u\right)=\frac{1}{k}\sum_{i=1}^{k}\max_{i\neq j}\left\{\frac{\Delta \left(X_{i}\right)+\Delta \left(X_{j}\right)}{\left(\delta \left(X_{i},X_{j}\right)\right)}\right\}$$
for $\Delta (X_{i})$ being an average distance between the points within a $X_{i}$ cluster ($\Delta (X_{j})$, respectively, for cluster $X_{j}$). As can be seen from the definition, the *Davies–Bouldin* index determines the average similarity between each cluster and the cluster closest to it. The clustering process becomes more complicated when the data contain outliers. It is then much more difficult to form internally coherent and externally separable clusters. However, when we want to detect outliers in data, clustering algorithms are probably best. If the clustered data contain outliers, the quality of the created clusters decreases significantly. Outliers do not fit into the groups.

## 4. Outlier Detection

This section presents a definition of outliers in data in the context of literature in contrast with our definition.

### 4.1. Outlier’S Definition

Outlier detection is finding data points that behave very differently from what is expected [12,13]. An outlier is an object in a data set that deviates significantly from the remaining data, has values far from the estimated or average values, or is not similar to any other object in its characteristics. The definition says, “An outlier is an observation that is far removed from the rest of the observations” [14]. An observation in a data set is called an outlier if at least one of the following conditions is met:It deviates from standard or known data behavior;It has values that are far from estimated or average values;It is not related or similar to any other element in the group in terms of its characteristics.

Outliers can contain valuable data about abnormal parameters of systems. Recognizing such non-standard parameters provides valuable information with specific applications. Some examples are as follows: earth science, medical diagnosis, intrusion detection systems, prevention of credit-card fraud [12]. One of the main problems is that no single scoring method would assess the similarity of two data points and how much they differ from each other in the data set.

If the data set contains outliers, we may have data preprocessing or statistical analysis errors. Any outlier in the data set can skew the test results and lead to an erroneous interpretation of the data. Thus, the removal of outliers is an essential task in the analysis and processing of data. Analyzing algorithms and models for removing outliers is of particular interest to scientists. Because not every outlier object must be an error in the data, we assume in our research that we do not remove the discovered outliers but only skip them in the further analysis-e.g., clustering. We introduce the outliers to the domain experts to look at these outliers closer.

### 4.2. Outlier Detection Algorithms

The issues of outliers detection in data are broadly analyzed in the literature. The methods proposed so far can be allocated to one of four groups: statistical-based, cluster-based, distance-based, and density-based methods.

The statistical-based outlier detection method assumes that the data have a specific regular distribution and we use the probability distribution to find out the data which deviate from the statistical distribution curve. Such data is an outlier when an occurrence probability is lower than a threshold value. It is necessary to know the characteristics of the data in advance to select a suitable distribution model. However, in practical applications, the data is unknown and complex. The data is mainly multi-dimensional. The time complexity of this method is very high, so it is not suitable for high-dimensional data.

The cluster-based detection method detects objects that do not belong to any cluster or a small cluster. This method focuses on the overall distribution of the data and performs outlier detection after clustering a data set. A certain number of clusters is formed and the clusters whose data points are significantly smaller than other clusters constitute an output in the form of outliers.

The concept of distance-based outlier definition is based on the following assumption: by calculating the distance from an object to its neighbors and sorting, the object with the largest value in the order is marked as an outlier.

Density-based outlier detection is proposed to overcome the shortcomings of distance-based detection of global outliers. The most known density-based method in the literature is the Local Outlier Factor (LOF) algorithm.

Local Outlier Factor (LOF) algorithm uses a density-based approach [15]. We detect anomalies by measuring a local density deviation at a given data point concerning the data points near it. We calculate a local density for all objects in the data set. We can identify data points with the same density as their neighbors and ones with a lower density by comparing the calculated density. Those with lower density are considered outliers. Density-based approaches distinguish the following two parameters that define the concept of density:The MinPts parameter (minimum points) indicating a minimum number of points;A parameter Eps defining a considered volume.

These parameters allow determining a density threshold for the algorithm, which decides whether or not a particular point is an outlier.

The idea of the LOF algorithm follows the 5 steps:In the first step an *Euclidean* distance between each pair of objects is calculated.In the next step, we calculate a distance dist_k(o) between a given data point *o* and its *k*-th nearest neighbor, using a so called *reachability-distance*.In step 3, for each data point *o* the *k*-distance neighborhood of *o* is calculated.In step 4 it is necessary to calculate reachability distances to all *k*-nearest neighbors of a point in order to determine a local reachability density of that point, which is computed by back-calculating the sum of all reachable distances of all *k*-nearest neighbors.Finally, in step 5, it is necessary to calculate an LOF for every data point.

LOF values are sorted and the highest $LOF_{k}\left(o\right)$ value is selected as possible outlier.

The Connectivity-based Outlier Factor (COF) algorithm is a variation of the LOF algorithm. The distinction lies in a different approach to assessing the density of cluster objects. The algorithm assigns a degree of outlier value to each data point. Unlike the LOF, the COF algorithm calculates *k*-nearest neighbors (k−NN) using a chain distance. This approach is based on the location of the data points. The objects in the cluster have a linear distribution. Chain distances represent the minimum total distance (between the first and last data points). Objects with high COF values are considered outliers.

In order to determine the COF value for each data point, we execute the following process:At first we find *k* nearest neighbors of the data point *o*. For each data point *o* we find the $N_{k}\left(o\right)$ set of its *k* nearest neighbors.Then we need to find a closest set-based path (SBN), which is an ordered sequence of *k* nearest data points starting with the point under consideration.Next it is necessary to find the cost of an SBN trail. We represent the trail as a set of weights of the respective edges.We consider the weight of an edge to be a distance between the two data points.After that we need to find an average chaining distance of the data point and finally a COF value of the data point.In the last step the COF values are sorted and the highest COF(o) value is selected.

### 4.3. The Concept of Outlier Detection Based on the Lof and Cof Algorithms-Our Approach

In our research, we want to check how the occurrence of outliers affects the quality of clusters. We assume that the quality of clusters with outliers is worse than without these outliers. Thus, it should be evident that we first examine the quality of the clusters with outliers, then ignore the identified outliers and redetermine the quality of the formed clusters. We will use the most known algorithms in the literature to discover the outliers: the LOF and the COF algorithms. Usually, the LOF and COF algorithms result in the same outliers detected.

The scheme of our approach is shown in Figure 1.

Input data undergoes necessary data preprocessing operations and then clustering (we choose the AHC or k−Means) algorithm. Then we evaluate the quality of clusters obtained in this way. We detect outliers (select the LOF or COF algorithm) and return them as one of the two elements constituting the data’s output. Bypassing the previously detected outliers, we cluster the input dataset again (and return the created clusters as the second of two elements of the output data) and assess the quality of the resulting clusters. We can compare the quality of data clusters containing outliers (A) with the quality of the clusters of the same input data without outliers (B). We expect that by excluding the outliers from the input dataset the quality of created clusters will increase.

## 5. Experiments

The experiments aimed at checking the impact of the clustering algorithms, clustering methods, and the selected distance measures on the effectiveness of outlier detection, measured by the response of cluster quality assessment indexes to remove outliers from the set. We wanted to see if the clustering algorithms and the outlier detection algorithms contributed similarly to improving the quality of clusters after detecting and removing outliers. We performed experiments on three different real datasets. We modified the number of detected outliers three times, using 1%, 5%, and 10% of the entire dataset as the number of outliers. We wanted to recognize the differences in the results. Our goal was to answer the question ”is it true that the more outliers we discover the better the quality of clusters without selected outliers will be”. In other words, we assume that if we discover outliers first and then cluster the data excluding the outliers, the quality of such clusters will be better than if we cluster the data including outliers. We analyzed two indexes for cluster quality assessment: the *Dunn* and the *Davies–Bouldin*. We measured the quality of clusters for the original data set in which potential deviations may occur. Then we look for outliers and omit them in the clustering process. In this way, we can compare the quality of clusters before and after removing outliers. Improving the quality of a cluster occurs if, after removing outliers, the quality measured by the *Dunn* index will increase as compared to the quality of clusters in which the outliers were not omitted in the clustering process. It is the completely opposite when the *Davies–Bouldin* index is concerned. Quality improvements occur when the quality of clusters measured by the *Davies–Bouldin* index decreases after removing outliers. Based on the experiments’ results, we can count in how many cases, after removing the outliers, the quality of the clusters has improved (i.e., the Dunn index increased, and the Davies–Bouldin index decreased).

### 5.1. Data Description

The source of the databases is the UCI Machine Learning Repository [16], a collection of databases and data generators used by the machine learning community to analyze machine learning algorithms empirically. A brief description of each data set is shown in Table 1.

The created databases differ in the number of instances and attributes and the types of attributes. *A* (Absenteeism at work Dataset) is the database created with absenteeism records from July 2007 to July 2010 at a courier company in Brazil. The set contains 740 instances, each consisting of 21 numeric (Integer, Real) attributes [17]. *B* (Shill Bidding Dataset) contains information about bidders, auctions, bids, prices, and auction duration. This dataset contains 6321 instances, each consisting of 13 mixed numeric attributes [18]. C (MoCap Hand Postures Dataset) a dataset containing 78,095 instances, with each instance consisting of 38 numeric (Integer, Real) attributes [19]. To record 12 users performing five hand gestures with markers attached to a left-handed glove, a Vicon motion capture camera system was used. It is worth mentioning that the dataset *B* originally was of mixed type. Only one feature was qualitative, but this feature has only one value, and we decided to exclude it in this analysis. Therefore, finally, all datasets were numeric. Qualitative data research will be the basis of our research in the future.

### 5.2. Methodology

The purpose of the experiments was to compare various clustering methods, clustering algorithms, and distance measures, which makes it possible to determine how changes in these parameters affect the final clustering results and how much the quality of outlier detection is improved. The steps involved in the experiments are described below. For each of the three datasets, the following experiments were carried out:Loading the dataset and preparing it correctly before applying clustering algorithms: preprocessing data using standarization, normalization, etc.;Data clustering using two different algorithms: k−Means with various number of clusters and AHC with different clustering methods (single, complete, average) and two different ways of measuring distance (*Euclidean* and *Chebyshev*). The tests were carried out with a different number of clusters in the range of *k*. Iteratively, starting with i=1 and increasing an *i*-th parameter by one at each step, the number of clusters *k* is calculated as $k\approx \sqrt{N}\pm i\%N$ until the condition that k≥2 and k<N is satisfied;Assessing the clustering quality using the *Dunn* and the *Davies–Bouldin* indexes;Finding 1%, 5%, and 10% of all outliers in the dataset using the LOF and COF. Removing the selected outliers and reclustering and recalculating the quality of clusters.

In total, we performed 686 experiments which are presented in this paper. The number of 686 experiments comes from the following calculation:
There are two clustering algorithms: k−Means and AHC;In case of AHC algorithm we may set the following values of the distance measures and clustering methods. For distance measure we have two options: *Euclidean* and *Chebyshev* distance measures. From the clustering methods we may choose one of three methods: single linkage (SL), complete linkage (CL), and average linkage (AL). Thus, using the AHC clustering algorithm we have 6 different combinations of given input parameters (see Figure 2).We adjust the number of created clusters to the size of the dataset. It means that for three used datasets we have various number of clusters. We do not want to check every possible value of *k* parameter because this would not be an efficient solution. The classical k−Means clustering algorithm requires multiple repetitions pf the clustering process for an iteratively changed (most often by 1) the number of clusters, starting from the value k=2. For a large data set, this process would be very ineffective. In the literature we can also come across an idea to divide the dataset into $\sqrt{N}$ of clusters. In our case, for the *A* dataset containing 740 of objects it would be 27 of clusters. Our idea is to adjust the number of different test values of the *k* parameter proportionally to the size of the analyzed data sets. Instead of that we propose to change the value of *k* iteratively according to the following formula. Starting with i=1 and increasing it by one at each step, the number of clusters *k* is calculated as
(6)$$k\approx \sqrt{N}\pm i\%N$$
until the condition that k≥2 and k<N is met. The calculated values of *k* parameter are included in Table 2.

For example, in case of the *A* dataset the calculation of *k* will be following:
-For i=1$k=\sqrt{740}\pm 1\%\cdot 740=34$ and 19;-For i=2$k=\sqrt{740}\pm 2\%\cdot 740=42$ and 12;-For i=3$k=\sqrt{740}\pm 3\%\cdot 740=49$ and 5;-For i=4$k=\sqrt{740}\pm 4\%\cdot 740=56$ (here we can not continue the process of calculationg *k* values because we met the stop criteria which is in this case k≥2 and k<N).

This solution will allow us to check different *k* parameter values adapted to the size of the input dataset.
The number of experiments is 686 as there are 8 versions of *k* parameter for the *A* dataset, 4 versions for the *B* dataset and 2 versions of *k* for the *C* dataset. We have 14 versions, and we repeat them for each of 6 different concepts of the AHC algorithm and 1 version of the k−Means algorithm. Adding all these combinations together, we reach 98 experiments.Choosing two outlier detection algorithms LOF and COF accordingly and for each of the three different variants of the number of outliers 1%, 5% and 10% we obtain the final number of experiments equal to 686.Every experiment contains the value of clustering quality indexes *Dunn* and *Davies–Bouldin* which are essential for comparing before and after excluding potential outliers from a given dataset.

### 5.3. Experimental Environment

To analyze clustering algorithms before and after removing the outliers, the Spyder programming environment (Python 3.8) was used, as well as the following libraries: Pandas for data processing and analysis [20], NumPy to perform basic operations on n-arrays and matrices: addition, subtraction, division, multiplication, transposition, calculating determinant, etc. [21], PyCaret to prepare the data for modeling, create an unsupervised anomaly detector, and prepare the model for predictions on unseen data [22] and Scikit-learn, one of the most widely used Python packages for data science and machine learning, which allows many operations and provides a great variety of algorithms for data processing, reduction in dimensions, model selection, regression, classification and cluster analysis [23].

The algorithms described in Section 3 and Section 4 have been implemented using Python and tested on the datasets described herein. We use Python 3.8 and the Anaconda package in this work, which includes many of the libraries required to run machine learning models, data mining, and output data in various formats. Existing Scikit-learn library models were used to implement the AHC and K−Means clustering algorithms, the *Dunn* and *Davies–Bouldin* indexes and the Pycaret library to implement the outlier detection algorithm. The program operates in the following way:Import Python analytical libraries Scikit-learn, NumPy, Pandas, PyCaret, and libraries to perform operations related to time.Implementation of algorithms:(a)AHC (algorithm_of_clustering) with parameters: *k* denoting the selected number of clusters, *linkage* denoting the type of linkage used in clustering, *affinity* denoting distance measures;(b)K−Means (kmeans) with a *k* parameter denoting the selected number of clusters;(c)*Dunn* and *Davies–Bouldin* algorithms (dunn_validator, davies_validator);(d)LOF and COF algorithms with parameter percent denoting the percentage of removed outliers.Data preparation functions:(a)df.replace—a function to replace the missing values with other values dynamically;(b)df.fillna—a function to replace Null values in Pandas data frame;(c)_normalize_databases—a function to normalize and standardize values in the data frame.Uploading and reading all three datasets.Execution of AHC, K−Means, LOF, COF, *Dunn*, and *Davies–Bouldin* algorithms on datasets.Transfering results to the Excel file.

### 5.4. Results

First, the impact of the percentage of detected outliers for both the LOF and COF algorithms was examined with regard to a frequency of improvement in the quality of clusters after removing the detected outliers. The results are presented in Table 3. We can see that using the *Davies–Bouldin* index was much likelier to improve the quality of clusters than the *Dunn* index, regardless of how many outliers were detected. It is essential to explain that all results presented in this Section are the average values of the analyzed parameters for each of the 686 experiments performed in this research.

All experiments present the number of cases in which there has been improvement, deterioration, or no changes in the values of the quality of clusters. The percentage values we see in the tables do not mean to represent an average value but the exact number of cases reflecting the event. It is expressed in percentage compared to all experiments from a given group. For example, in Table 4, we can see that when 1% of outliers are discovered and removed (there are 196 such cases), in 129 of these 196 cases, which is 65.82%, the quality of clusters measured by the *Davies–Buldin* index has improved. In 102 cases in this group, the quality of clusters measured by the *Dunn* index has improved. The Tables are extended by a piece of additional information (the number of cases confirming a given event).

Then we decided to check whether any of the clustering algorithms used contributed more to improving the quality of clusters than the other after removing outliers. The results are presented in Table 4.

It turns out that taking into account all the experiments performed, the quality of the clusters was higher after removing the outliers, more often for the k−Means algorithm than for AHC. Furthermore, this is regardless of whether the *Dunn* or *Davies–Bouldin* index was used. We see that not all the differences studied are statistically significant. At the level of statistical significance, p<0.05, we will say that in the case of the *Davies–Bouldin* index, the use of the K−Means algorithm for clustering data has much more often led to a record improvement in the quality of clusters after removing deviations. In other words, the K−Means algorithm is not resistant to the presence of outliers. Therefore, no statistically significant differences in the quality of clusters were noticed when we eliminated outliers using the *Dunn* index.

An important task was to examine the impact of using the outlier detection method on the frequency of improvement of the quality of clusters after removing outliers. Table 5 contains the results. There is an interesting tendency there.

Using the COF outlier detection algorithm the increase in quality of created clusters is achieved much more often than using the LOF algorithm. It means that COF algorithm depends more significantly on the occurence of outliers. We notice that using the COF algorithm statistically significantly (p<0,05, ${\mathrm{Chi}}^{2}$ Pearson Test) more often leads to improving the quality of clusters after eliminating the outliers. Therefore, the COF algorithm tends to discover more significant outliers. After removing them, the quality of the clusters improves.

We also wanted to check if and how the distance measures contribute to improving the quality of the clusters. It turns out that when using the *Euclidean* distance measure, the improvement of cluster quality is more often achieved for the *Davies–Bouldin* index, while for the *Chebyshev* measure, the quality of the clusters is more often improved by using the *Dunn* index. As Table 6 indicates, there are no statistically significant differences (p>0.05) in the effectiveness of improving the quality of clustering after removing outliers depending on what distance measure (*Euclidean* or *Chebyshev*) we use.

Knowing that the analyzed datasets are real datasets that differ with respect to the size and type of the analyzed data, we also decided to investigate the differences in the frequency of increase or decrease in clustering quality depending on the input data source. The results are presented in Table 7.

Types of data sets we analyze significantly impact how effective the process of outlier detection is and consequently impact the quality of the created clusters. There are statistically significant differences for each of the analyzed datasets in the frequency of improvement in the quality of clusters after removing previously found outliers.

Table 8 also presents interesting results. We can see that depending on which set was analyzed, the quality of clusters did not constantly improve as the number of detected deviations increased. It is also impossible to unequivocally determine whether any of the measured indexes of the quality of clusters always allows obtaining an improvement in the quality of clusters. This confirms that the size and type of analyzed data have a significant impact on the effectiveness of deviation detection and the quality of clustering.

We see a trend in which the more deviations we detect and turn off from clustering, the more often the quality of the clusters improves. We should point out that in the end, the analyzed dataset with a specific type of data determines the effectiveness of outlier detection and improves the quality of clusters.

The last analyzed clustering parameter, which can affect the improvement of the quality of clusters after removing the outliers, is the cluster combinination method. Table 9 indicates that there are statistically significant differences between the clustering methods (single, complete, average) in the frequency of improvement of cluster quality after removing outliers. We can see that outliers removal improves the quality of clusters by less than 30 percent of cases (using the *Dunn* index to assess the quality) while using the single method. In the case of the complete or average method, this effect is obtained much more often (about 80%).

### 5.5. Discussion

The research concludes that the COF algorithm more often improves the quality of clusters than LOF by removing the outliers. In the context of clustering algorithms, the K−Means algorithm reacts much more actively to the outlier’s presence and skipping. Probably the reason is that this algorithm is much less resistant to the appearance of outliers in the set than the hierarchical algorithm. The research also confirmed the original assumption that the more outliers we remove from the set, the better the quality of the clusters would be. However, an essential conclusion seems to be that the input data type significantly affects the results achieved: the quality of the clusters created for data containing potential outliers.

## 6. Summary

In this research, we assessed the influence of the clustering parameters, the clustering algorithms, and the outliers detection methods on the quality of created clusters. Several hundred experiments were performed, where individual clustering parameters (distance measures, number of clusters, clustering algorithms) and outliers detection parameters (number of outliers and outlier detection algorithm) changed for three different data sets. We checked which factors responded positively to outliers. It turned out that a vast majority of experiments confirmed the thesis that if a data set contains outliers, it will negatively affect the quality of created clusters. Therefore, this should prompt us to search for outliers before clustering large and real data sets. By removing the outliers first, we will be able to form good-quality clusters from the data and, therefore, achieve a greater efficiency in exploring such datasets. An additional benefit of outlier detection will be a reduced clustering time (as there is no longer any difficulty in a cluster formation). Consequently, a better quality of the created clusters will translate into improved quality of explored knowledge. Of course, the detected outliers are, by definition, passed on to field experts who have a chance to explore knowledge in a previously underexplored area.

## Figures and Tables

**Figure 1 entropy-24-00917-f001:**
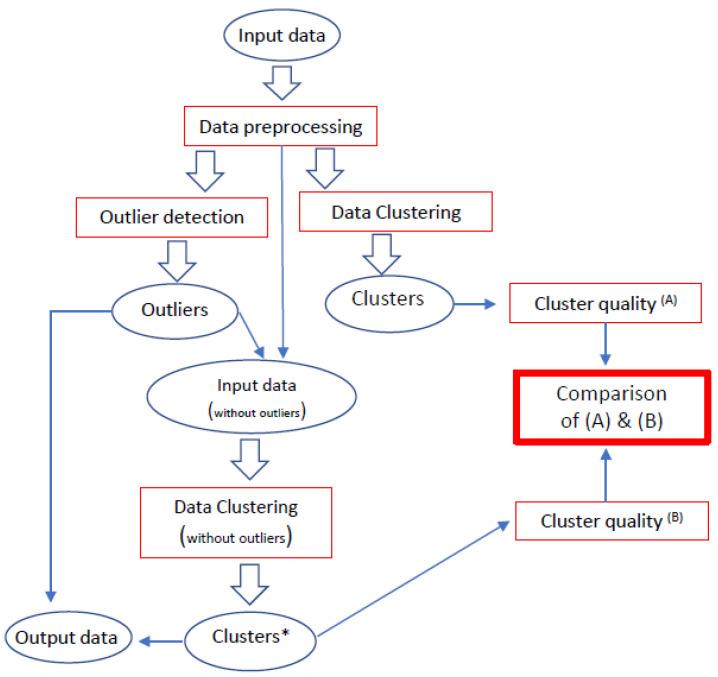
The concept of outlier detection based on LOF and COF algorithms—our approach. Clusters * mean clusters after removing discovered outliers.

**Figure 2 entropy-24-00917-f002:**
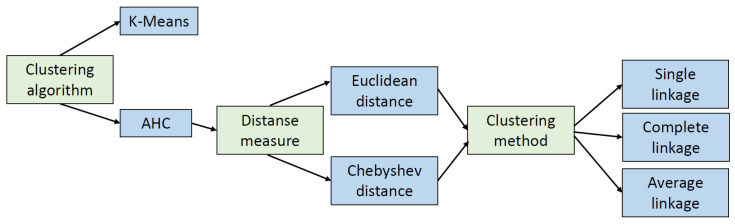
Clustering parameters used in the research.

**Table 1 entropy-24-00917-t001:** Description of the data sets.

Dataset	Type	Number of Instances	Number of Features
A	numeric	740	21
B	mixed	6321	13
C	numeric	78,095 (First 12,000 used in experiments)	38

**Table 2 entropy-24-00917-t002:** Description of the databases.

Dataset	*N*	*k* Values
A	740	5, 12, 19, 27, 34, 42, 49, 56
B	6321	16, 79, 142, 205
C	12,000	109, 229

**Table 3 entropy-24-00917-t003:** The impact of the number of outliers of the frequency of increase in cluster quality.

	Increase in Quality		Decrease in Quality		No Change
% Of Outliers/	Dunn	Davies–Bouldin	Dunn	Davies–Bouldin	Dunn
# Cases					
1%	52.04%	65.82%	32.14%	34.18%	15.82%
	102	129	63	67	31
5%	63.78%	78.57%	31.12%	21.43%	5.10%
	125	154	61	42	10
10%	67.86%	82.14%	32.14%	17.86%	0.00%
	133	161	63	35	0
Chi ${}^{2}$ Pearson		Dunn index: *p* = 0.00000		Davies–Bouldin index: *p* = 0.00136	

**Table 4 entropy-24-00917-t004:** The impact of the number of outliers of the frequency of increase in/decrease in the cluster quality.

	Increase in Quality		Decrease in Quality		No Change
Clustering	Dunn	Davies–Bouldin	Dunn	Davies–Bouldin	Dunn
Algorithm					
AHC	60.91%	73.81%	31.15%	26.19%	7.94%
	307	372	157	132	40
K−Means	63.10%	85.71%	35.71%	14.29%	1.19%
	53	71	30	12	1
Chi ${}^{2}$ Pearson		Dunn index: *p* = 0.07329		Davies–Bouldin index: *p* = 0.01882	

**Table 5 entropy-24-00917-t005:** LOF and COF algorithms for cluster quality indexes.

Outlier	Dunn			Davies–Bouldin	
Detection					
Algorithm	No Change	Increase	Decrease	Increase	Decrease
LOF	13.95%	53.40%	32.65%	68.03%	31.97%
	41	157	96	200	94
COF	0%	69.05%	30.95%	82.99%	17.01%
	0	203	91	244	50
Chi ${}^{2}$ Pearson		Dunn index: *p* = 0.00000		Davies–Bouldin index: *p* = 0.00014	

**Table 6 entropy-24-00917-t006:** Distance mesures for cluster quality indexes.

Distance	Dunn			Davies–Bouldin	
Measure	No Change	Increase	Decrease	Increase	Decrease
Euclidean	7.94%	58.33%	33.73%	76.19%	23.81%
	20	147	85	192	60
Chebyshev	7.94%	63.49%	28.57%	71.43%	28.57
	20	160	72	180	72
Chi ${}^{2}$ Pearson		Dunn index: *p* = 0.44332		Davies–Bouldin index: *p* = 0.22409	

**Table 7 entropy-24-00917-t007:** The frequency of improving the quality of clusters according to the type of data.

Dataset	Dunn			Davies–Bouldin	
	No Change	Increase	Decrease	Increase	Decrease
A	10.12%	60.71%	29.17%	78.57%	21.43%
	34	204	98	264	72
B	2.38%	60.12%	37.50%	74.40%	25.60%
	4	101	63	125	43
C	3.57%	65.48%	30.95%	65.48%	34.52%
	3	55	26	55	29
${\mathrm{Chi}}^{2}$ Pearson	Dunn index: *p* =0.00727			Davies–Bouldin index: *p* = 0.04103	

**Table 8 entropy-24-00917-t008:** The frequency of improving the quality of clusters according to the type of data and the number of discovered outliers.

	Increase in Clustering Quality Indexes					
	Dunn			Davies–Bouldin		
Dataset	1%	5%	10%	1%	5%	10%
A	43.75%	64.29%	74.11%	68.75%	83.04%	83.93%
	49	72	83	77	93	94
B	60.71%	62.50%	57.14%	64.29%	76.79%	82.14%
	34	35	32	36	43	46
C	67.86%	64.29%	64.29%	57.14%	64.29%	75.00%
	19	18	18	16	18	21
Chi ${}^{2}$ Pearson	Dunn index: *p* = 0.00000			Davies–Bouldin index: *p* = 0.00136		

**Table 9 entropy-24-00917-t009:** The frequency of improving the quality of clusters according to the clustering method.

Dataset	Dunn			Davies–Bouldin	
	No Change	Increase	Decrease	Increase	Decrease
Single	13.69%	29.76%	56.55%	58.33%	41.67%
	23	50	95	98	70
Complete	2.98%	79.76%	17.26%	88.10%	11.90%
	5	134	29	148	20
Average	7.14%	73.21%	19.64%	75.00%	25.00%
	12	123	33	126	42
${\mathrm{Chi}}^{2}$ Pearson	*p* = 0.00000			*p* = 0.00000	

## Data Availability

Not applicable.

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
