# Peer review of "How the Outliers Influence the Quality of Clustering?"

_entropy, 2022, doi:10.3390/e24070917_

Round 1

Reviewer 1 Report

The topic of the paper is quite interesting and indeed not covered in detail in the literature. I see, however, several  issues:

* The English should be thoroughly revised, there are a number of errors and some parts are difficult to follow.

* Numbered citations are used but they are not ordered, which is not correct for a scientific paper.

* My main concern is about the presentation of the results. Authors have a number of experiments, but summarise them just with a mean. This is clearly not enough to assess the results. Standard deviations can be shown, at least for the main conclusions. Even better, density plots would illustrate the results and improve the quality of the paper.

* Another weird thing is the fact that there are only numerical variables, but a dataset contain originally attributes. Moreover, the background sections are focused on numerical distances without mention dissimilarity measures. Maybe this could be stated at the beginning of the paper, and mention in the conclusions the study attribute data as future work.

* Real data sets are very interesting, but maybe in this case a simulation exercise controlling the generation of outliers (even artificial groups) would be also appropriate in order to check the hypothesis.

* Finally, I miss some hypothesis testing to demonstrate that the differences are significant regarding the improvement of the quality. In line with a previous comment, showing two sample means is just not enough to reach valid conclusions.

Author Response

We want to thank the reviewers for thoroughly assessing our work. We tried to consider every remark and made many changes to the article. We hope that our paper will meet with the acceptance of reviewers in the current form. Of course, we remain at the reviewers' disposal if they expect additional changes.

Reviewer 2 Report

The article touches on an interesting topic in terms of improving data clustering techniques.

Unfortunately, the style of presenting the material is more like a popular science article or a thesis than a scientific article.

The article has a rather strange structure. For example, sections 1, 2 and 3 are an introduction. They can be combined and reduced. These sections provide too much explanation of terminology, which can be omitted in a scientific article. In addition, it would be nice to confirm the information from section 3 with appropriate references. Is paragraph 1.1 needed in a scientific article?

In paragraph 2, the main difference between the cited articles and this study is only the "absence of outliers" is this true, perhaps these methods have other differences?

There is no logic in the presentation of the material in the whole article.

Unfortunately, this type of article cannot be published in the journal.

Author Response

(The authors gave the same response as above.)

Round 2

Reviewer 1 Report

Thank you for taking into account my suggestions, now the paper has a stronger foundation and the results are scientifically demonstrated.

Reviewer 2 Report

Thanks to the authors for their careful attention to the comments of the reviewers. In this form, the article fully reflects the study.